

# Characterizing the spatial distribution of coral reefs in the South-Central Coast region of Viet Nam using Planetscope imagery

Khanh V. Nguyen[1], Vinh C. Duong[2], Kinh T. Kieu[1], Thuong V. Tran[3], Cho-ying Huang[4], Ruth Reef[5] and Thien M. Hoang[1]

[1] Faculty of Biology and Environment, The University of Danang-University of Science and Education, Danang City, Viet Nam
[2] Department of Environment and Natural Recourses, Gia Lai Campus, Nong Lam University – Ho Chi Minh, Pleiku city, Gia Lai Province, Viet Nam
[3] Institute of Engineering and Technology, Thu Dau Mot University, Thu Dau Mot city, Binh Duong Province, Viet Nam
[4] Department of Geography, National Taiwan University, Taipei, Taiwan
[5] School of Earth, Atmosphere and Environment, Monash University, Clayton, Victoria, Australia

Corresponding author
Thuong V. Tran,
thuong.tran@tdmu.edu.vn

## ABSTRACT

This study aims to understand the spatial distribution of coral reefs in the central region of Viet Nam. We classified live coral cover in Son Tra Peninsula (ST) and Cu Lao Cham Island (CLC) in the South-Central Coast Region of Viet Nam using the Maximum Likelihood Classifier on 3 m Planetscope imagery. Confusion matrices and the accuracy of the classifier were assessed using field data (1,543 and 1,560 photographs in ST and CLC, respectively). The results showed that the reef's width ranged from 30 to 300 m across the study site, and we were able to detect live coral cover across a depth gradient of 2 to 6 m below the sea surface. The overall accuracies of the classifier (the Kappa coefficient) were 76.78% (0.76) and 78.08% (0.78) for ST and CLC, respectively. We found that 60.25% of coral reefs in ST were unhealthy and the live coral cover was less than 50%, while 25.75% and 11.46% of those in CLC were in good and excellent conditions, respectively. This study demonstrates the feasibility of utilizing Planetscope imagery to monitor shallow coral reefs of small islands at a high spatial resolution of 3 m. The results of this study provide valuable information for coral reef protection and conservation.

## INTRODUCTION

Coral reefs make up only 0.1% of the global ocean substrate, yet they are one of the most biodiverse marine ecosystems and they play a key role in providing a range of ecosystem services (*Hoegh-Guldberg, Pendleton & Kaup, 2019*; *Obura et al., 2019*; *Pawlik & McMurray, 2020*). Southeast Asia contains the largest area of coral reef cover on Earth, estimated at 91,700 km$^2$, followed by the Australia–Papua New Guinea region at 62,800 km$^2$

(*Guan et al., 2020*). Coral cover in the SE Asia region has declined and is being threatened by both local and global stressors such as marine pollution and runoff, direct destruction, overexploitation of key species, outbreaks of coral predators, and climate change (*Zhou et al., 2018*; *Hoegh-Guldberg, Pendleton & Kaup, 2019*; *Carriger, Yee & Fisher, 2021*). Long term monitoring programs of coral health in the SE Asia region are important for managing this global biodiversity hotspot which is facing a range of impacts (*Tun et al., 2008*), Characterize coral reef cover and health over large spatial and temporal scales is critical for the effective management and protection of coral reef resources (*McCarthy et al., 2017*; *Zhou et al., 2018*). The use of remote sensing to monitor reefs over large temporal and spatial scales will be necessary to provide standardized data for reef health across the SE Asian region over this period of rapid change and population growth.

Coral reef monitoring applies both quantitative measurements to explore area coverage of basic reef bottom-types (*e.g.*, coral cover), and provides information about coral reef health. These monitoring frameworks are designed to develop effective strategies for management, conservation, and restoration of coral reefs across various spatial scales (*Guan et al., 2020*; *Carriger, Yee & Fisher, 2021*). Coral coverage and health is conducted using two primary approaches including *in-situ* surveys and remotely sensed data (*Hedley et al., 2016*; *Hedley et al., 2018*; *McCarthy et al., 2017*; *Gonzalez-Rivero et al., 2020*). *In-situ* methods provide a high resolution images from a handle camera and compositionally detailed quantitative description of coral reef biodiversity, through SCUBA driver surveys (*e.g.*, underwater geo-located digital photography) or high resolution aerial images that are costly and time consuming (*Levy et al., 2018*; *Li et al., 2019*). Thus, while *in-situ* methods can provide unrivaled detail on species diversity, coral density, and individual colony size and health; the spatiotemporal scale of these analyses is limited and biased against remote reefs, periods of severe weather conditions, or challenging reef topography (*Levy et al., 2018*). Satellite remote sensing to acquire coral information at different scales has been a cost-effective alternative or complement for *in-situ* surveys (*Hedley et al., 2018*; *Levy et al., 2018*; *Zhou et al., 2018*; *Li et al., 2019*). Among various sensors, the medium spatial resolution (30 m) Landsat images has been the most widely used for regional mapping coarse coral habitat classes (*e.g.*, 04 classes) (*Tobler, 1988*; *Zhou et al., 2018*). Recently, freely available (for education and research programs) high spatial resolution (3 m) Planetscope imagery has demonstrated potential in coral pattern observation providing daily images (*Lemajic, Vajsová & Aastrand, 2018*; *Lazuardi, Wicaksono & Marfai, 2021*; *Mansaray et al., 2021*; *Wulandari & Wicaksono, 2021*).

Viet Nam has a large marine area with over 3,260 km of coastline and more than 3,000 islands (*Nguyen & Nguyen, 2014*). The distribution of coral reefs in Viet Nam is along the coast and around offshore islands, and is categorized into five separate regions (*i.e.*, western Tonkin Gulf, Middle-central, South-central, South-eastern, and South-western) (*Hedberg et al., 2017*). The highest coral diversity is in the South-Central area (SCA) with 400 coral species recorded (*i.e.*, Son Tra Peninsula (ST), Da Nang, Cu Lao Cham Island (CLC), Nha Trang, and Ninh Hai) (*Latypov & Selin, 2008*; *Nguyen & Vo, 2013*). However, the health of coral reefs (as defined by live coral cover) in the SCA has noticeably declined in the past few decades, with the condition of ∼50% of the reefs in this region currently

classified as poor/very poor and only 11.6% and 2.9% of the reefs classified as being in good and excellent conditions, respectively (*Nguyen & Phan, 2008*; *Nguyen, 2009*; *Nguyen & Vo, 2013*). Previous studies utilized various medium and high spatial resolutions satellite data such as Landsat, SPOT-5, GeoEye-1, IKONOS, QuickBird, and VNRedsat-1 to map general reef bottom-types, coral coverage, and reef building or expansion in this region (*Tran, Phinn & Roelfsema, 2012*; *Nguyen, Luong & Ho, 2015*; *Nguyen, Bui & Nguyen, 2019*; *Nguyen et al., 2019a*; *Nguyen et al., 2019c*). However, these studies have not examined the spatial distribution of reef health (live coral cover). Therefore, the objectives of this study were: (i) to utilize Planetscope imagery, a free satellite for education and research programs, for determining coral reef types at a high spatial resolution; (ii) to understand spatial distribution of coral reef categories (*i.e.*, live coral cover and classification of benthic categories) in ST and CLC in the South-Central area of Viet Nam.

## MATERIALS & METHODS

### Study sites

The study areas (*i.e.*, ST (108°13′58.1″E–16°08′54.4″N and 108°20′22.4″E–16°05′16.2″N) and CLC (108°24′42.8″E–15°58′57.7″N and 108°33′04.2″E–15° 53′58.4″N)) are nature reserve areas located in the South-Central coast region of Viet Nam (*Nguyen, Luong & Ho, 2015*; *Nguyen, Huynh & Zhang, 2015*; *Hoang et al., 2019*) (Fig. 1). CLC was recognized as a world biosphere reserve by UNESCO in 2009 (*Nguyen, Luong & Ho, 2015*; *Nguyen, Huynh & Zhang, 2015*). The study sites are located within the tropical monsoon region with a distinct dry season (January–July) receiving 600 mm rainfall over these 6 months and a rainy season, where 80% of the annual rainfall (2,000 mm) falls between the month of August to December. The annual mean temperature at the study areas is about 26 °C with little seasonal variation. The region is regularly disturbed by typhoons and river run-off in the wet season, which can lead to destruction of reefs.

The reefs at study area are shallow fringing reefs, with a diversity of 226 and 227 coral species in ST and CLC, respectively (*Kimura, Tun & Chou, 2014*; *Kimura, Tun & Chou, 2018*). Deposited materials on the ocean floor in this region is sand and rubble, which deposit between the inner and outer reef regions. Both regions are important to their local economies providing revenue from coral reef-based tourism, and small-scale fisheries. However, these reefs are under threat from an accumulation of stressors such as human disruption, typhoons, new outbreaks of crown of thorns starfish, excess terrigenous sediment output due to land use change, pollution, overfishing, destructive fishing, and the mining of reefs for limestone (*Nguyen & Vo, 2013*; *Quach, 2018*). A lack of regular monitoring has impacted on effective coral reef management and protection (*Le, 2020*).

### Data collection and Processing

The Planetscope images (3 m spatial resolution) used in this study were obtained from the Planet Education and Research Program server (https://www.planet.com/markets/education-and-research/) (Table 1). The wavelengths of Planetscope imagery cover the blue (455–515 nm), green (500–590 nm), red (590–670 nm), and near-infrared (780–860 nm) spectral regions (*Gabr, Ahmed & Marmoush, 2020*). These images were acquired on
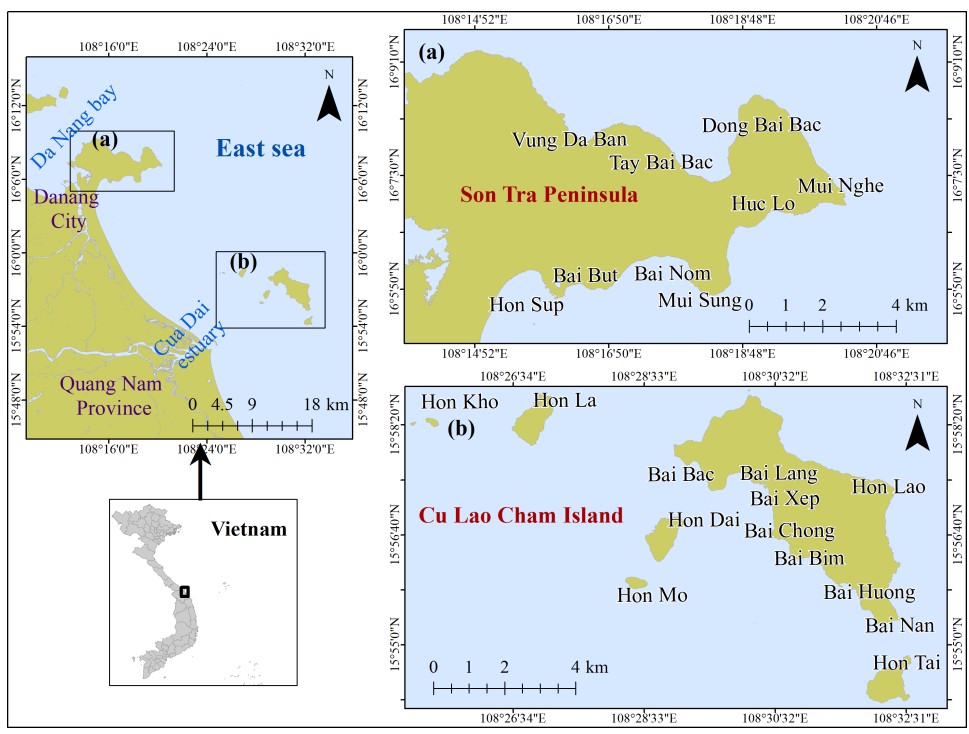

**Figure 1** Location of study area (A) Son Tra Peninsula (ST) and (B) Cu Lao Cham Island (CLC).

**Table 1** Details of the acquired Planetscope imagery.

| Location | Row | Path | Data of acquisition | Spatial resolution |
|---|---|---|---|---|
| Son Tra | 021 | 858 | 28 June 2019 | |
| | | 859 | | 3 m |
| Cu Lao Cham | 025 | 351 | 08 August 2019 | |
| | | 352 | | |

28 June 2019 (ST) and 08 August 2019 (CLC) with 0% of cloud cover and coordinate system projection of WGS 84/ UTM zone 49N. The images, acquired by a Dove satellite with additional postprocessing applied, known as the PlanetScope Ortho Scene Product (level 3B), were ortho-rectified, atmospherically corrected, and scaled to surface reflectance (*Lemajic, Vajsová & Aastrand, 2018*).

We carried out field surveys during August and September 2019 to gather information about the depth and features of the ocean bottom. The bottom features were measured following the 25's method, proposed by (*Roelfsema, Phinn & Joyce, 2006*), in a grid of 5 m × 5 m using an underwater camera (GoPro HERO 8, USA). Based on this approach, five photographs including the four corners and one center positions of a 1 m × 1 m quadrat were taken. The underwater photographs were ortho-rectified to control the spatial uncertainty to <3 m in order to match the resolution of the satellite data. The depth and the coordinates of each photograph were recorded using a Hondex PS-7FL

(Honda Electronics Ltd., Toyohashi, Japan) and Garmin GPS MAP 62sc (Garmin Ltd., Lenexa, Kansas, USA), respectively. In addition to photo capture in the 5 m × 5 m grid, underwater 360 video transects (CND713, CND Co., Ltd., Korea) were taken to determine the ocean floor classes (Table 2) in deep and homogenous areas before collecting substrate types (*i.e.,* sand, and hard bottom) to reduce the cost of field survey. A total of 1,543 photographs and 167 video transects at ST and 1,560 photographs and 174 video transects at CLC were collected during the field campaign. The photos were filtered to select for good quality photographs demonstrating their ability to clearly identify the bottom feature. Five photographs were overlapped to create a transect and each transect was geo-referenced based on captured coordinates before analyzing the benthic composition by the Coral Point Count Excel (CPCe) software (*Kohler & Gill, 2006*; *Roelfsema, Phinn & Joyce, 2006*).

Bottom types, derived from the photographs of 1 m × 1 m quadrats, were identified into four classes (*i.e.,* live coral [hard and soft coral], sand, hard bottom [rock, rubble, dead coral], deep water (>14 m)) (*Hill & Wilkinson, 2004*), and were used to calculate proportion of substrate composition. The coral reef health in 5 m × 5 m grids was categorized as the 'poor', 'fair', 'good' or 'excellent' condition based on live coral cover (*Chou et al., 1995*; *Hill & Wilkinson, 2004*) (Table 2). Finally, a part of the ground-truth samples was randomly selected for image classification training (see the sub-section in Supervised classification and coral cover mapping). A total of 323 and 369 randomly selected ground-truth samples in the ST and CLC, respectively, were identified to assess classification accuracy. It included each classed category as follow: 1LC (69), 2LC (64), 3LC (47), HB (43), S (51), and D (49) in Son Tra and 1LC (62), 2LC (56), 3LC (54), 4LC (55), HB (43), S (55), and D (44) in Cu Lao Cham. (Table 2 and the Supplemental Files).

## Sun glint correction and water column correction

Sun glint occurs in images under conditions of clear skies, shallow waters (when wind generated waves disturb the surface), more so when images are acquired at a high spatial resolution (*Hedley, Harborne & Mumby, 2005*). This phenomenon could be defined as the specular reflection of light from the water surface, when the component of sensor-received radiance due to the surface reflection is greater than the water-leaving radiance from sub-surface features (*Kay, Hedley & Lavender, 2009*). Previous studies proposed sun-glint removal methods for ocean color applications with resolutions on the scale of 100–1,000 m (*Wang & Bailey, 2001*; *Hedley, Harborne & Mumby, 2005*; *Kay, Hedley & Lavender, 2009*). However, these models could only correct moderate glint and large errors remain in the brightest glint areas (*Kay, Hedley & Lavender, 2009*). Hence, a separate set of revised methods were developed for benthic remotely sensed data at high spatial resolutions (<10 m) (*Hochberg, Andréfouët & Tyler, 2003*; *Kay, Hedley & Lavender, 2009*). These approaches use data from the near-infrared (NIR) to determine the amount of glint in the received signal (*Hedley, Harborne & Mumby, 2005*; *Kay, Hedley & Lavender, 2009*), thus improving data retrieval for bathymetry or coral habitat classification (*Hochberg, Andréfouët & Tyler, 2003*; *Hedley, Harborne & Mumby, 2005*; *Kay, Hedley & Lavender, 2009*). In this study, the sun light correction of the Planetscope images were undertaken using a revised 'de-glinting' method (*Hochberg, Andréfouët & Tyler, 2003*; *Hedley, Harborne & Mumby, 2005*).

**Table 2 Identification scheme of bottom types and coral reef health.**

| Class | Detail | Class description | Coral reef health criteria | Label | Depth range (m) | Sample image |
|-------|--------|-------------------|---------------------------|-------|-----------------|--------------|
| | | $0 < LC \leq 25\%$ | Poor | 1LC | ST: 0.8–8.5<br>CLC: 0.6–11.5 |  |
| | | $25 < LC \leq 50\%$ | Fair | 2LC | ST: 0.5–8.2<br>CLC: 0.9–9.9 |  |
| | | $50 < LC \leq 75\%$ | Good | 3LC | ST: 1.3–7<br>CLC: 0.8–10.2 |  |
| Live coral (LC) | hard and soft coral | $LC > 75\%$ | Excellent | 4LC | CLC: 0.9–10.5 |  |
| Sand (S) | Sand dominated | $LC = 0\%$ and $S \geq 50\%$ S | | S | ST: 0.5–13.1<br>CLC: 0.4–12.5 |  |
| Hard bottom | Rock, rubble, dead structure | $LC = 0\%$ LC and $HB > 50\%$ | | HB | ST: 1–8.2<br>CLC: 0.7–10.5 |  |
| Deep zone | | Depth > 14 m | | D | ST: 14–24<br>CLC: 14–29.5 |  |
Following this approach, the relationships between NIR and visible bands were explored by linear regression based on a sample of the image pixels. The pixel sample was collected from one or more regions of the image where a range of sun glint is evident, but where the underlying spectral brightness would be expected to be consistent (normally deep-water areas) (*Hedley, Harborne & Mumby, 2005*). If the slope of the regression line for band i is $b_i$, then all pixels in the image could be de-glinted in band i, following Eq. (1):

$$L_i^{'} = L_i + b_i(L_{NIR} - Min_{NIR}) \tag{1}$$

where, $L'_i$ is the de-glinted pixel of visible band i; $L_i$ is reflectance from the visible band i; $b_i$ is the regression slope; $L_{NIR}$ is reflectance from the NIR band; and $Min_{NIR}$ is the minimum value of the NIR band.

In addition to the sun glint phenomenon, we needed to correct for the water column so that bottom classes were not misclassified due to different water column conditions (*Zoffoli, Frouin & Kampel, 2014*). Among various algorithms, Lyzenga's equation (*Lyzenga, 1978*; *Lyzenga, 1981*) is widely applied to empirically diminish water column attenuation impacts. This method enhances mapping accuracy in digital classification processes of bottom types, and does not require knowing the local depth across every pixel of the scene (*Green et al., 2000*; *Pahlevan, Valadanzouj & Alimohamadi, 2006*; *Pu et al., 2012*; *Zoffoli, Frouin & Kampel, 2014*). The algorithm uses multispectral image datasets to generate 'depth –invariant' coefficients of bottom types from each pair of spectral bands. The result of the correction creates multiple 'depth - invariant bands, replacing original bands to classify bottom features (*Green et al., 2000*). The depth-independent composition of corrected radiance ($DII_{ij}$) in bands i and j (pseudo-color band) was generated by Eq. (2):

$$DII_{ij} = \ln(L_i) - \left[\left(\frac{k_i}{k_j}\right)\ln(L_j)\right] \tag{2}$$

where $L_i$ is de-glinted pixel of band i; $L_j$ is de-glinted pixel of band j; $\left(\frac{k_i}{k_j}\right)$ is the ratio of attenuation coefficients.

In this study, three visible bands of PlanetScope imagery with high water penetration were plotted against transformed band pairs (*i.e.,* band 1 and band 2, band 2 and band 3, band 1 and band 3). The slope values of the ratio of attenuation coefficients for each band were applied to generate three DII bands (*i.e.,* $DII_{12}$, $DII_{13,}$ and $DII_{23}$) to class the bottom types.

## Supervised classification and coral cover mapping

Different types of supervised classification methods have been used to identify coral reefs with water column corrected images and bottom classes in similar marine regions. The Maximum Likelihood Classifier was used due to its effectiveness and adaptability, compared to other classifiers (*Ali, Qazi & Aslam, 2018*; *Cabral et al., 2018*). In the current study, the Maximum Likelihood Classifier was subsequently applied to each image. This classifier was based on a probability density function and guessed the probability with which a specific pixel belongs to a specific category (*Richards, 1999*).The standard false colour satellite images were composed before applying the supervised classification approach.

**Table 3  Interpretation of Kappa values.**

| Value | Description |
|---|---|
| Less than 0 | Less than chance agreement |
| 0.01–0.2 | Sigh agreement |
| 0.21–0.40 | Fair agreement |
| 0.41–0.60 | Moderate agreement |
| 0.61–0.80 | Substantial agreement |
| 0.81–1.00 | Almost perfect agreement |

The standard implementation of supervised maximum likelihood classification required training samples representing the feature types to be classified (*Lillesand, Kiefer & Chipman, 2015*). Taking this *a-priori* knowledge into account, each pixel was labelled as 1LC, 2LC, 3LC, 4LC, HB, S and D (Table 2). Finally, we used a 3× 3 focal majority filter to minimize the "salt-and-pepper effect" in the coral reef maps (*Booth & Oldfield, 1989*).

Finally, to validate the classification results, an error matrix was produced to assess the performance. We used independent training sites to assess the accuracy of each classification. The error matrix showed seven classes using four metrics: overall accuracy (OA), producer's accuracy (PA), user's accuracy (UA), and the Cohen Kappa coefficient. The Kappa coefficient was computed using Eq. (3):

$$K = \frac{Observed\ Accuracy - Change\ Agreement}{1 - Change\ Agreement} \tag{3}$$

The Kappa coefficient measures the agreement between the classification and ground-truth values. A Kappa value of 1 represents perfect agreement, while a value of 0 represents no agreement. We interpreted the Kappa values as per Table 3 following (*McHugh, 2012*; *Nguyen, Luong & Ho, 2015*; *Nguyen, Huynh & Zhang, 2015*). All the image processing and post-classification steps, presented in Fig. 2, were processed using ArcGIS v. 10.8 software packages (ESRI, USA).

# RESULTS

## Descriptive field data analysis

The correlation coefficients and ratio of attenuation coefficients among visible bands at good water penetration (Fig. 3). To perform the water column correction, homogenous sand substrates of varying depths collected from field surveys were applied to calculate the ratio of attenuation coefficients. The very high correlation coefficients of 0.95 above were explored between bands 1 and 2, while the relationship between bands 1 and 3 showed high $R^2$ values of 0.85 in both ST and CLC.

In addition to the water column correction, the study analyzed the spectral signatures among the bottom types (*i.e.,* coral lives and non-coral classes) based on 04 visible bands after corrected sun glint. Many studies measured the spectral characteristics from *in situ* data or from remotely sensed data (*Green et al., 2000*; *Hochberg, Andréfouët & Tyler, 2003*; *Yamano & Tamura, 2004*; *Leiper et al., 2014*; *Giardino et al., 2019*). These studies considered that separating spectral features among coral reef bottom classes is challenging.

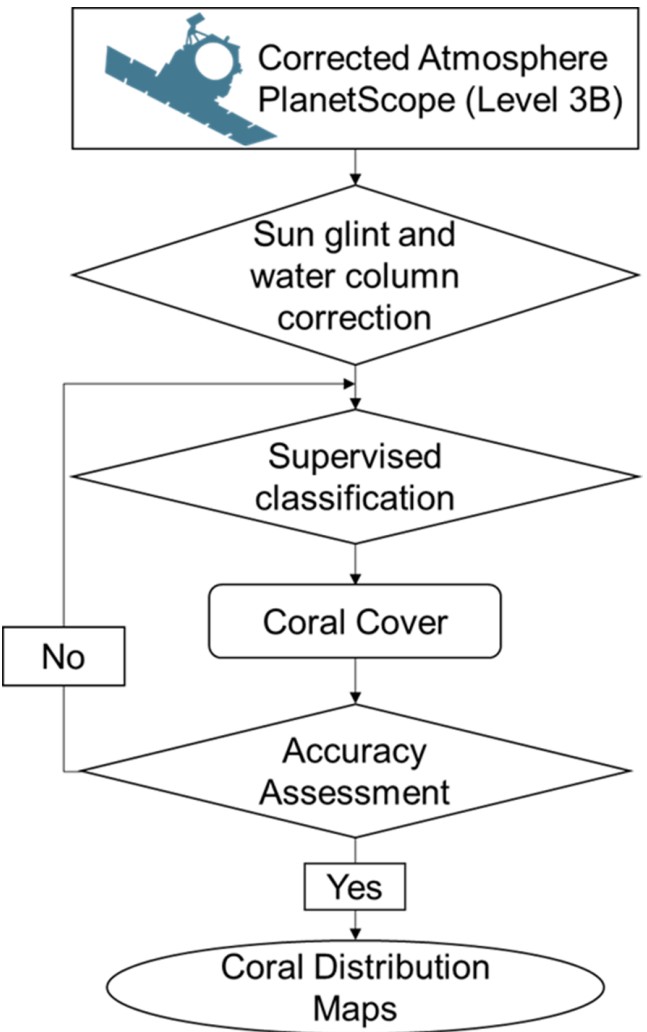

**Figure 2 Flowchart of coral distribution mapping in the study area.**

The classes with good separation are sand dominated *vs* deep water classes representing light and dark objects, respectively (*Green et al., 2000*; *Hochberg, Andréfouët & Tyler, 2003*). Our analysis shows that good discriminating ability is also found between the D and S classes, while HB bottom easily overlapped with 1LC and 2LC classes in the blue and green bands in both study areas. For live corals, the spectral characteristics utilized a separation among bands blue, red, and NIR (Fig. 4). The reflectance of bottom types among these classes decreased in the blue to red bands while increasing at the NIR band. Therefore, the high separation among bands permits the classification of live coral and bottom types in both areas.

## The patterns of coral reefs distribution
From the analysis of field survey results, the most common coral reef types in the study areas were fringing reefs, reefs directly attached to the shoreline of the peninsula and the

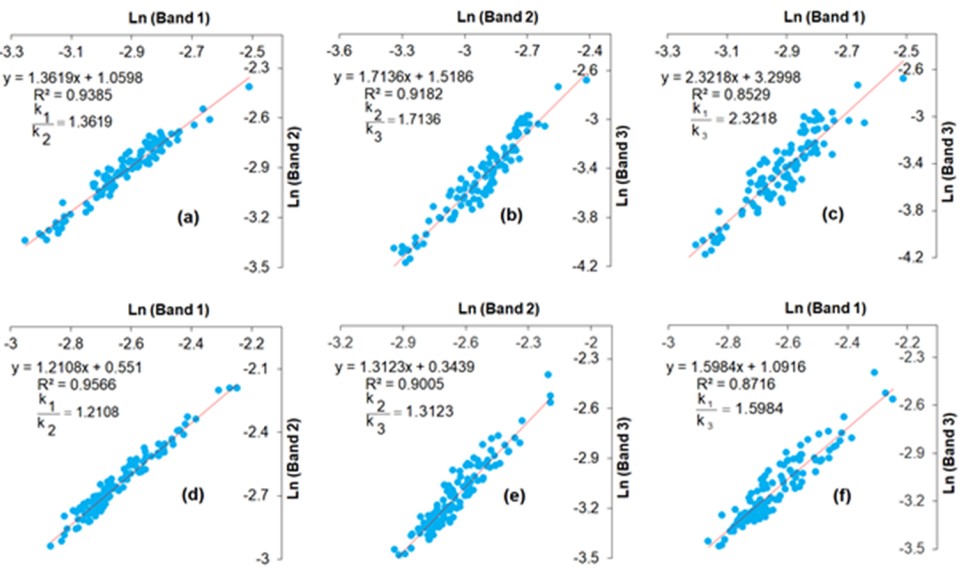

**Figure 3** Transformed band ratios of Planetscope bands in ST ((A) bands 1 and 2, (B) bands 2 and 3, and (C) bands 1 and 3) and in CLC ((D) bands 1 and 2 (E) bands 2 and 3, and (F) bands 1 and 3).

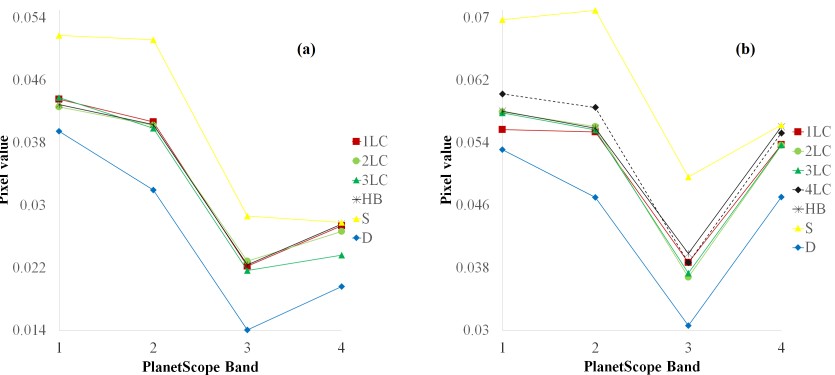

**Figure 4** Spectral signatures of (A) six classes at ST and (B) seven classes at CLC.

islands. This reef type varied in relation to a combination of coastline geomorphology and seabed characteristics, normally ranging between 30 m and 300 m in width. The vertical pattern of coral distribution in the study area reached to 14 m in depth and was densest between 2 and 6 m (Fig. 5).

The depth distribution of coral reefs and other substrates is presented in Figs. 6 and 7 for ST and CLC, respectively. Coral reefs in ST were located along the shoreline from Tien Sa in the northwest to Hon Sup in the southwest. The northwestern most coral reefs in this region were not included in this study due to the high turbidity of the Han estuary. High coral cover was found in the inner coral reefs of the northern ST, while coral cover was patchier in the southern region as shown by the alternating distribution of three classes (1LC, 2LC and 3LC; poor, fair, and good coral cover respectively) across the reef (Fig. 6).

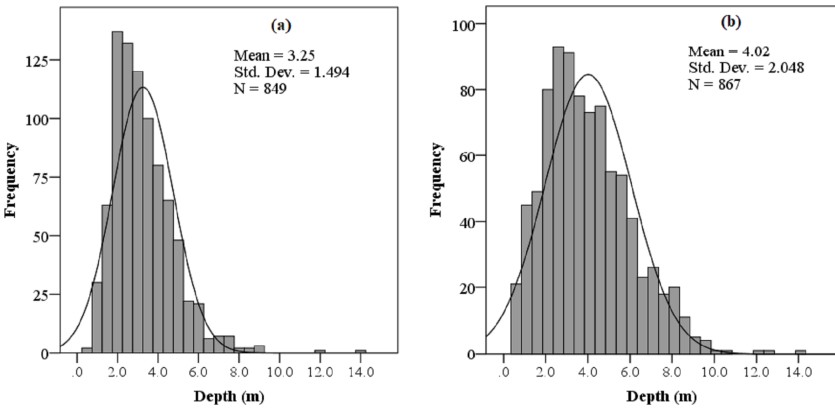

**Figure 5** Depth histogram of coral distribution in (A) Son Tra Peninsula and (B) Cu Lao Cham Island.

In southwest ST including Bai Nom, Bai But, and Hon Sup (Figs. 6D–6E), the reef width ranged between 140–300 m, while across the rest of ST reefs were narrower (30–90 m wide).

In CLC, coral reefs were located throughout the western and northern shores of Hon Lao (main island) and around small islands (*i.e.,* Hon Kho, Hon La, Hon Mo, Hon Tai, and Hon Dai) (Figs. 7A, 7B, 7G, and 7H). High coral cover was found predominantly in the southern part of the smaller islands, while the northern shores of the islands recorded lower coral cover (Fig. 7). The widest reef (150 m) was the SunGroup reef (near the tourist area managed by SunGroup) while the reef width at the rest of the island locations ranged between 50–100 m.

Following the classification of the bottom cover into different categories (Table 2), we estimated that the live coral areas were 0.47 $km^2$ and 0.58 $km^2$ in the ST and CLC, respectively. Bottom cover proportion is reported in Table 4. In ST, the health condition of ranged from 'poor' to 'fair condition' with 60.25% classified as a coral reef in poor condition (1LC), 31.88% in 'fair condition' (2LC), and only 7.87% in 'good condition' (3LC). Several regions (*e.g.,* Vung Da Ban and Tay Bai Bac - Dong Bai Bac) recorded over 70% of coral cover in poor condition, while none of the reefs were in good condition or above.

Compared to ST, the health condition of coral reefs in CLC was better and more consistent across reefs. Despite the better condition, the percentage of reefs in poor health still accounted for 34.28%, followed by 2LC and 3LC at 28.51% and 25.75%, respectively. However, the proportion of coral reefs in 'excellent condition' in the CLC was 11.46%, compared to none in the ST. This is perhaps due to several programs to protect and recover coral reefs in the CLC (*Hua et al., 2015*; *Quach, 2018*) such as the reef recovery program at some regions of Cu Lao Cham Island (*e.g.,* Hon Tai, Hon La and Hon Dai, Bai Bac and Bai Huong), which have contributed to coral reef health (*Hua et al., 2015*).

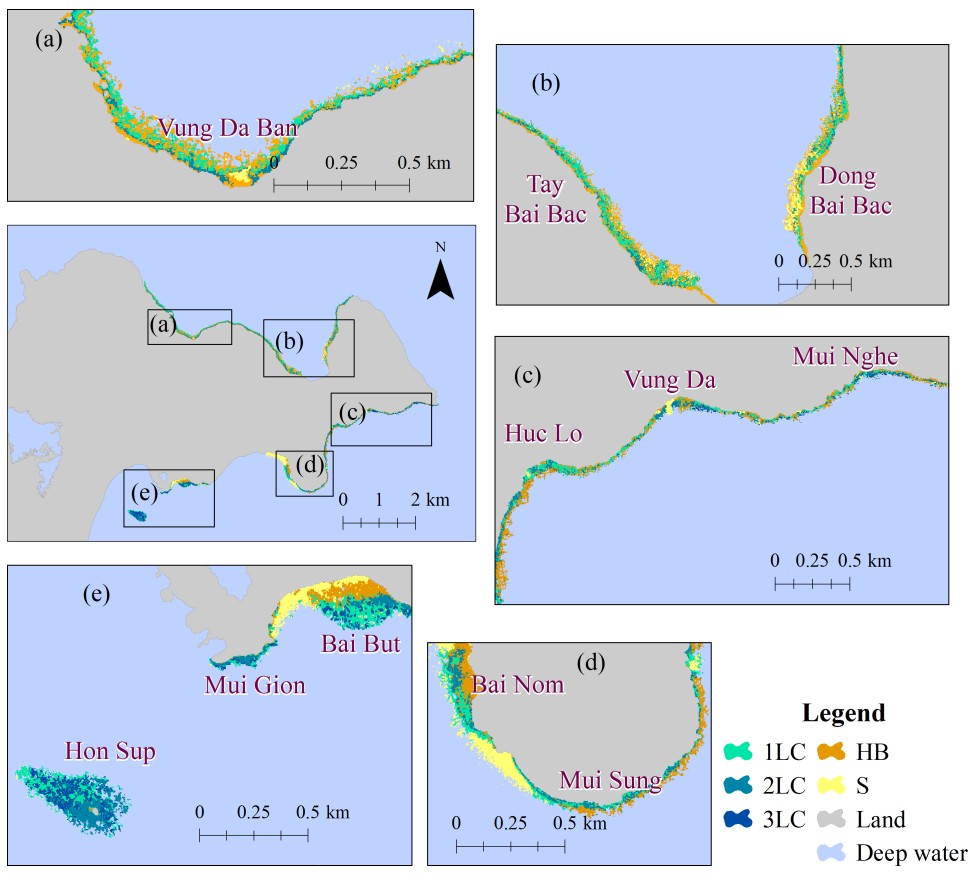

**Figure 6** (A–D) Coral cover in Son Tra Peninsula. 1LC: Poor; 2LC: Fair; 3LC: Good; S: Sand; and HB: Hard bottom.

## Performance assessment

The Planetscope images provided three visible spectral bands that enabled us to penetrate to the water bottom and perform the classification of the benthos (*Green et al., 2000*; *Mumby et al., 2004*) and a NIR band that we used for sun-glint removal, subsequently improving the classification accuracy (*Hedley, Harborne & Mumby, 2005*; *Anggoro, Siregar & Agus, 2016*). Assessments of accuracy were carried out on the coral cover and bottom classes by comparison with geo-rectified ground reference data collected during field surveys. Error matrices were generated for each of the classified images in ST and CLC (Tables 5 and 6, respectively). The overall classification accuracies (Kappa coefficients) for ST and CLC were 78.08% (0.78) and 76.78% (0.76), respectively. Among the various coral and bottom types, deep zone and sands class showed the highest producer's accuracies (PA) and user's accuracy (UA) in both regions because of both the exponential decrease of light intensity with increasing depth and the high light absorption through the column before reaching to the deep bottom (*Green et al., 2000*; *Goodman, Purkis & Phinn, 2013*). In the coral cover classification, the highest PA (78.57%) and UA (80.85%) were found for 2LC and 3LC, respectively in ST, followed by the 1LC at 76.56% and 71.01%, respectively. Whereas these

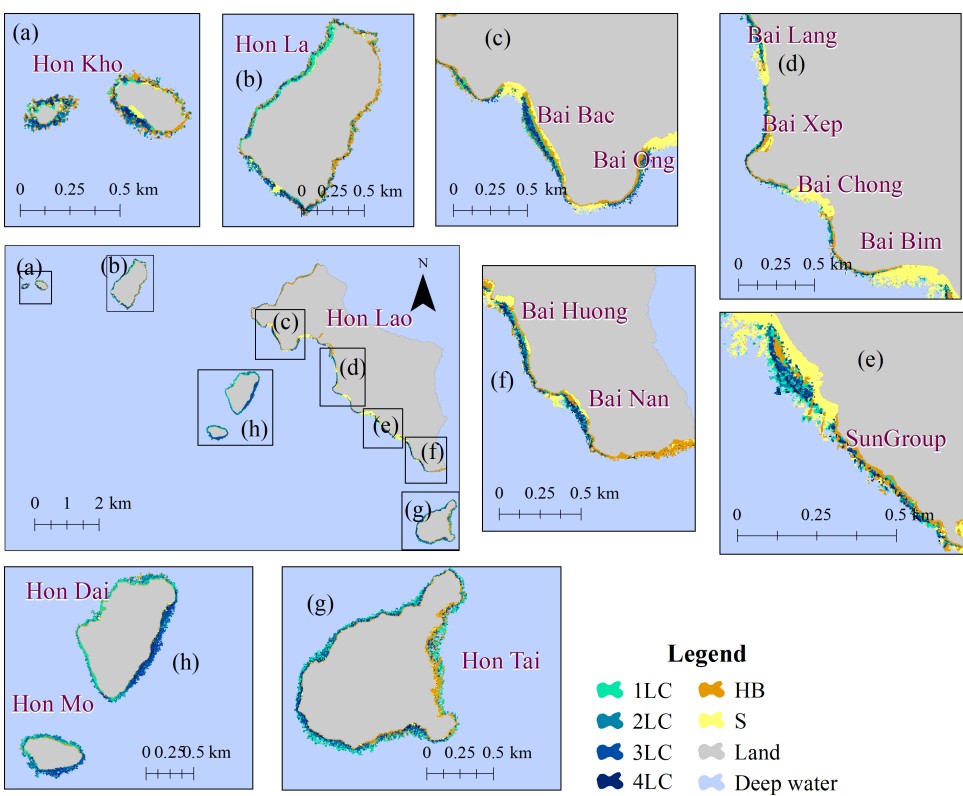

**Figure 7** (A–G) Coral cover in Cu Lao Cham Island. 1LC: Poor; 2LC: Fair; 3LC: Good; 4LC: Excellent; S: Sand; and HB: Hard bottom.

**Table 4** Percentage of coral cover classes in the coral reefs of Son Tra Peninsula and Cu Lao Cham Island.

|  | Level (%) Location | 1LC | 2LC | 3LC | 4LC |
|---|---|---|---|---|---|
| Son Tra | Vung Da Ban | 75.19 | 24.81 | – | – |
|  | Tay Bai Bac - Dong Bai Bac | 71.05 | 28.95 | – | – |
|  | Mui Nghe –Huc Lo | 48.68 | 38.06 | 13.26 | – |
|  | Mui Sung –Bai Nom | 59.48 | 40.52 | – | – |
|  | Bai But –Hon Sup | 46.51 | 33.76 | 19.73 | – |
|  | **Total** | **60.25** | **31.88** | **7.87** | **–** |
| Cu Lao Cham | Hon Kho | 20.15 | 36.26 | 30.42 | 13.17 |
|  | Hon La | 42.08 | 28.21 | 22.67 | 7.04 |
|  | North of main island (NI) to Bai Ong | 15.54 | 54.31 | 25.76 | 4.39 |
|  | Harbour –Bai Bim | 32.87 | 38.29 | 21.46 | 7.38 |
|  | SunGroup | 45.1 | 13.49 | 25.26 | 16.15 |
|  | Bai Huong –Bai Nan | 33.54 | 19.15 | 25.98 | 21.33 |
|  | Hon Tai | 46.26 | 19.7 | 26.81 | 7.23 |
|  | Hon Mo –Hon Dai | 31.59 | 23.78 | 27.07 | 17.56 |
|  | **Total** | **34.28** | **28.51** | **25.75** | **11.46** |

**Table 5  Confusion matrix of bottom types in Son Tra Peninsula.**

| Class | 1LC | 2LC | 3LC | HB | S | D | Total | UA |
|---|---|---|---|---|---|---|---|---|
| 1LC | **49** | 5 | 3 | 7 | 3 | 2 | 69 | **71.01** |
| 2LC | 5 | **44** | 10 | 3 | 1 | 1 | 64 | **68.75** |
| 3LC | 1 | 7 | **38** | 1 | 0 | 0 | 47 | **80.85** |
| HB | 6 | 0 | 0 | **32** | 3 | 2 | 43 | **74.42** |
| S | 1 | 0 | 0 | 3 | **41** | 6 | 51 | **80.39** |
| D | 2 | 0 | 1 | 1 | 1 | **44** | 49 | **89.8** |
| Total | 64 | 56 | 52 | 47 | 49 | 55 | **OV** | 76.78 |
| PA | 76.56 | 78.57 | 73.08 | 68.09 | 83.67 | 80 | **KC** | 0.76 |

**Table 6  Confusion matrix of bottom types in Cu Lao Cham Island.**

| Class | 1LC | 2LC | 3LC | 4LC | HB | S | D | Total | UA |
|---|---|---|---|---|---|---|---|---|---|
| 1LC | **47** | 7 | 1 | 3 | 2 | 1 | 1 | 62 | **75.81** |
| 2LC | 3 | **46** | 5 | 2 | 0 | 0 | 0 | 56 | **82.14** |
| 3LC | 0 | 7 | **43** | 3 | 1 | 0 | 0 | 54 | **79.63** |
| 4LC | 0 | 4 | 5 | **40** | 6 | 0 | 0 | 55 | **72.73** |
| HB | 5 | 4 | 2 | 0 | **30** | 1 | 1 | 43 | **69.77** |
| S | 0 | 2 | 0 | 0 | 6 | **43** | 4 | 55 | **78.18** |
| D | 1 | 2 | 0 | 0 | 2 | 0 | **39** | 44 | **88.64** |
| Total | 56 | 72 | 56 | 48 | 47 | 45 | 45 | **OV** | 78.05 |
| PA | 84 | 63.89 | 76.79 | 83.33 | 63.83 | 95.56 | 86.67 | **KC** | 0.78 |

indicators were quite different in CLC. Major areas of confusion and misinterpretation were found to roughly coincide with the transitional areas between hard bottom (HB) and coral classes.

## DISCUSSION

Coral reef health in almost regions of Viet Nam is declining due to land use conversion at coastal and marine areas under the pressure of economic development (*Nguyen & Nguyen, 2014*; *Nguyen et al., 2019a*; *Nguyen et al., 2019b*). Monitoring and detecting coral reefs plays a crucial role in proposing natural resource management policy, environmental management, and conservation efforts (*Asner, Martin & Mascaro, 2017*). Using remote sensing tools can address the coral reef health knowledge gap for much of the region. However, the performance remotely sensed image classification depends on the relationship between the spatial resolution of remotely sensed data and the number of categories in the classification (*Mumby & Edwards, 2002*; *Capolsini et al., 2003*; *Wicaksono & Lazuardi, 2018*; *Li et al., 2019*). For example, *Wicaksono & Lazuardi (2018)* presented a range of Kappa values of 0.21–0.40 by using Planetscope (3 m) for mapping benthic habitat and seagrass species in Karimunjawa Islands with five classes, while a higher Kappa value of 0.81 was found in *Li et al. (2019)* by using Dove (3 m) satellite data to map coral reef habitats in marine the sanctuary Arrecifes del Sureste. Therefore, the selection of adaptive data for coral observation at the region scale was highlighted in this study. The visualization of

PlanetScope data for mapping the spatial distribution of benthic community has also been confirmed by recent studies in Viet Nam (*Nguyen et al., 2019b*). The first study in Viet Nam using Planetscope images to monitor submerged aquatic vegetation (5 classes) showed a high overall accuracy and Kappa coefficient at 92.52% and 0.90, respectively (*Nguyen et al., 2019a*; *Nguyen et al., 2019b*; *Nguyen et al., 2019c*). Therefore, with 07 classification categories, the Kappa coefficients and the overall accuracy in our study indicate sufficient accuracy for post classification assessment.

In addition to Planetscope images, previous studies applied other sensors to explore coral distribution in Viet Nam. For example, *Tran, Phinn & Roelfsema (2012)* used images of IKONOS (3.2 m), QuickBird (2.4 m), and GeoEye-1 (5 m) for categorizing benthic cover. More recently, *Nguyen, Luong & Ho (2015)* and *Nguyen et al. (2019b)* applied medium resolution satellite images [e.g., Landsat 8 OLI (30m), SPOT-5 (10m)] for coral mapping in Ly Son island and Nam Yet island, respectively. However, these studies were only characterized coral cover to other benthic cover and bottom types and unclassified or divided coral lives into separate class. Hence, findings in this study are one of the very first studies applying the Planetscope imagery for monitoring coral cover patterns with 04 coral live classes in the South-Central Coast of Viet Nam. Besides, the outcomes in our study were comparable to those of *Capolsini et al. (2003)* and *Phinn, Roelfsema & Mumby (2012)* that used commercial Quickbird-2 (2.4 m), WorldView-2 (0.5 m), and IKONOS (3.2 m) images, respectively to explore coral reefs. Here we affirm the potential of Planetscope imagery for visualizing higher resolution coral health monitoring metrics. This approach is expected to provide managers with more accurate data, reduce time, manpower and cost in marine conservation in developing countries.

We mapped coral distribution in ST and CLC as a case study of the South-Central Coast region in Viet Nam. However, some limitations have been discovered. Offsets associated with the difference between the actual positions and field locations when a scuba diver took underwater photos and GPS estimates from the surface reduced spatial certainty to 5 m × 5 m. Hence, around deep-water areas (15 m depth) especially where the bottom morphology is narrow, each photograph covered an area over 5 m$^2$, reducing the number of field images for bottom feature identification. Besides, around the field sites with homogenous bottom characteristics (*i.e.,* sand, hard bottom, and deep water), individual photos and the camera of 360 CND713 were used to record the bottom, to identify neighboring areas, and to enhance the ground truth data for the D class. Therefore, the number of filtered and random ground truth data to perform error matrix for post classification in this study reduced.

For future research, a time-series of PlanetScope images should be obtained and applied to explore the spatiotemporal pattern of coral reef variation. Besides, intersection between coral pattern and various information layers (*e.g.*, geology, climatic condition, biomes, land use and land cover change) should be employed to better understand distribution and diversity and to propose strategies for coral conversation. Nonetheless, optimizing the data collection and analysis related to classification and field survey methods should be compared to enhance accuracy assessment.

## CONCLUSIONS

The distribution of coral reefs (coral cover and bottom types) in ST and CLC in the South-Central Coast of Viet Nam was characterized using very high spatial resolution (3 m) Planetscope imagery with a supervised classification approach. Seven classes, including coral reef health (04 classes) and bottom types (03 classes), were categorized to determine coral cover and condition of the reefs in the study area. Field surveys of 1,543 photographs in ST and 1,560 photographs in CLC were used to assess accuracy.

The study characterized coral reefs distribution and health in a peninsula setting and offshore islands. The results revealed that the overall condition of coral reef was of 'poor condition' in the Son Tra (60.25%) and ranged between 'fair and good condition' in CLC. Additionally, the reefs in both areas were predominantly fringing reef, reaching a depth of 14 m and a width of 300 m. In ST, there were better environmental conditions in the southern area than in the northern area which was indicated by the higher density of live coral in Hon Sup, Mui Nghe and Mui Gion. In CLC, there was higher coral cover in the south such as in Bai Nan, Bai Huong, Hon Dai and SunGroup than in the north and center of both small and large islands, and Hon Tai in the south.

Accuracy assessment of the classified images using Maximum Likelihood classifier showed satisfactory performance and a high accuracy in both areas with an overall accuracy of 75% and Kappa coefficients of >0.75. Therefore, the overall approach provides an accessible means for reef scientists and managers to effectively apply remotely sensed data to observe coral ecosystems. The findings of this study can provide a comprehensive understanding of coral pattern and support policymakers/planners to develop appropriate coral reefs conservation policies in coastal regions.

## ACKNOWLEDGEMENTS

The authors would like to thank the Planet Education and Research Program for supporting the Planetscope images in this research.

### Funding

This research was funded by the Ministry of Education and Training, grant number B2019-DNA-04 under the project ''Applying GIS and remote sensing to understand coral reef conservation in the South Central, Viet Nam''. The funders had no role in study design, data collection and analysis, decision to publish, or preparation of the manuscript.

### Grant Disclosures

The following grant information was disclosed by the authors:
The Ministry of Education and Training: B2019-DNA-04.

### Competing Interests

Cho-ying Huang is an Academic Editor for PeerJ.

## Author Contributions

- Khanh V. Nguyen conceived and designed the experiments, performed the experiments, prepared figures and/or tables, and approved the final draft.
- Vinh C. Duong conceived and designed the experiments, performed the experiments, analyzed the data, prepared figures and/or tables, and approved the final draft.
- Kinh T. Kieu performed the experiments, authored or reviewed drafts of the paper, and approved the final draft.
- Thuong V. Tran analyzed the data, prepared figures and/or tables, authored or reviewed drafts of the paper, and approved the final draft.
- Cho-ying Huang and Ruth Reef analyzed the data, authored or reviewed drafts of the paper, and approved the final draft.
- Thien M. Hoang performed the experiments, prepared figures and/or tables, and approved the final draft.

## Data Availability

The raw data is available in the Supplementary Files.

## Supplemental Information

Supplemental information for this article can be found online at http://dx.doi.org/10.7717/peerj.12413#supplemental-information.

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
