# Peer review of "Characterizing the spatial distribution of coral reefs in the South-Central Coast region of Viet Nam using Planetscope imagery"

_PeerJ, doi:10.7717/peerj.12413_

## Round 0.1 · original submission · Major Revisions

According to reviewer 2, the manuscript needs to be restructured, and the novelty of the research shall be clarified and emphasized. Therefore, a major revision is required.

Reviewer 1 ·

Basic reporting

No comment

Experimental design

No comment

Validity of the findings

No comment

Additional comments

The article is well written. I have a few minor comments

Line 33: I believe PlanetScope is a commercial satellite. Therefore, it is not clear what you mean by “public domain”
Line 34: The phrase “high spatial resolution” is a relative term. High in reference to what? For example, the 3 m resolution of PlanetScope may not be as high a resolution as for a satellite that has a spatial resolution of 0.5 m.
Line 56: Please explain what high resolution means in this context.
Line 57: Define SCUBA
Line 68: Please check if there is a 5-m resolution PlanetScope. I think it should be 3-m only. The 5-m resolution should be RapidEye.
Line 87: Please check if PlanetScope is free. As far as I know, it is a commercial satellite. Nonetheless, it has free imagery of 10,000 km2 per month. Depending on the size of the area you are looking at, this could be enough to monitor coral reef in your study area.
Line 255: I suggest that you use a consistent unit (use km2 instead of ha).

Reviewer 2 ·

Basic reporting

- The article does not provide clear progress on how the PlanetScope image was use to mapping coral reefs.
- Several sentences and information have not been cited (line 72).
- The satellite remote sensing images were acquired on 28 June 2019 (ST) and 08 August 2019 (CLC) with 0% of cloud cover and coordinate system projection. However, August is the beginning month of raining season in central Vietnam, so the impacts of rain, typhoon, river run-off would be the significant effects on water quality and water transparency. The authors should include these negative impacts on the manuscript.
- The structure of the article should conform to an acceptable format of ‘standard sections’ (see our Instructions for Authors for our suggested format). Significant departures in structure should be made only if they significantly improve clarity or conform to a discipline-specific custom.
- Figures should be relevant to the content of the article, of sufficient resolution, and appropriately described and labeled.
- All appropriate raw data have been made available in accordance with our Data Sharing policy.
- The mapping results need to consider because did not provide any spectral difference on each classes (1LC, 2LC, 3LC, 4LC, HB, S and D).
- The validation results need to be rechecked, especially the Kappa Coefficients

Experimental design

The submission should clearly define the research question, which must be relevant and meaningful. The knowledge gap being investigated should be identified, and statements should be made as to how the study contributes to filling that gap.

The author should add the number of sample site observations used in the survey (sample bottoms). For example: 47 1LC, 20 2LC, 30 HB, 40 S, 50D,…
I wonder that nothing has been said about atmospheric correction during processing. I think dust, gas, aerosols, and air molecules in the atmosphere could affect surface.
The author can add more information about the Ki/Kj ratio of the PlanetScope band pair and reflectance spectrum correlation coefficients for each band pair (R2)
This study use OA, PA, UA, Kappa to assess accuracy. Therefore, total different accuracy measures are four, not seven (line 214).
All Figures: please provide each map a letter a-c and clarify these in the caption. It would be good to add a scale bar and scale text (ratio) and a north arrow for each of the maps. It is also required the map frame to present the coordination of the study areas.
Table 1: Class description is not clear. LC > 0-25% or Cover of LC from 0-25%??? I suggest rewrite this part.
Table 1: The author should add the depth of live coral, Sand, Hard bottom. I suggest rewrite this part.

Methods were not described with sufficient information.
The author can add more information about wavelengths and radiometric resolution characteristics of PlanetScope.
Line 130-131: It is not sure the authors used 25’s method in a grid of 5 x 5 m using underwater camera.
Line 134: Provide more details of accuracy and GPS model of Garmin GPS unit (USA).
Line 136-137: A total of 1,543 photographs in ST and 1,560 photographs in CLC were collected. Please provide more details about the photo locations and captured days. The depth of captured photos.
Need to describe more details about accuracy assessment. What is the cited references?

In the methodology section, it is presented 1,543 photographs in ST and 1,560 photographs in CLC were collected and used for accuracy assessment. However, the Table 3 – Confusion matrix of bottom types in Son Tra Peninsula and Table 4 - Confusion matrix of bottom types in Cu Lao Cham Island were presented very few number of points. This is a huge flaw that requires further work to address.

Moreover, the Abstract section presents “The overall accuracies of the classifier (the Kappa coeficient) were 78.08% (0.78) and 76.78% (0.76%) for ST and CLC, respectively”. This conclusion is also make a remarkable flaw as in the Table 3 and Table 4 present the results in the reverse figures.

Validity of the findings

The novelty of the work is unclear. Methods use for determining coral reef types at a higher spatial resolution are not new, and understand spatial distribution of coral reef categories have been quantified in many studies. If the knowledge gap is site specific, it would suffice for local reporting while for an international journal, a more global knowledge gap needs to be addressed.

The paper seems interesting. However, the novelty of your research is relatively low in the journal study to be considered as a research article. As the research methods were not descripbed clearly and used very traditional/ old methods, lower-performing parametric classifier, none robust contribution.

The findings of this study can provide a comprehensive understanding of coral pattern and support policymakers/planners to develop appropriate coral reefs conservation policies in coastal regions.

Discussion and results should be divided into two part. I suggest rewrite this part.
The author should compare the accuracy of the classification to other areas of Vietnam.
The main potential problem of this manuscript is the local framing of the research, which preset as a case study.
The methods were not clearly and correctly presented. Further improvements to the methods are strongly required. The results were not inconsistent, sufficient robust and poteintial flaw.
I think it could be published if the authors could improve the manuscript and framing their work in international literature and comparing to others reported in the literatures.

Additional comments

- This manuscript is not well prepared.
- The manuscript should be rewritten, hence, discussion and results should be divided into two separated parts.
- The author should compare the accuracy of the classification to other areas of Vietnam.
- The main potential problem of this manuscript is the local framing of the research, which preset as a case study.
- The methods were not clearly and correctly presented. Further improvements to the methods are strongly required. The results were not inconsistent, sufficient robust and potential flaw.

I think it could be published if the authors could improve the manuscript and framing their work in international literature and comparing to others reported in the literatures.

---

## Round 0.2 · Minor Revisions

Please check the the annotated document for more editing suggestions.

Reviewer 1 ·

Basic reporting

The authors have committed to addressing my comment.

Experimental design

The authors have committed to addressing my comment.

Validity of the findings

The authors have committed to addressing my comment.

Additional comments

The authors have committed to addressing my comment.

Reviewer 2 ·

Basic reporting

It is fine in revised version

Experimental design

Also good, authors added some information about methods

Validity of the findings

Yes, it is clarified

Additional comments

The manuscript has been significantly improved. However, there are few minor things must be changed:
- the country name must be Viet Nam, it is right words as Ministry of Home Affairs of Viet Nam
- Vietnamese references must be changed, eg, from 563-568, 665-677

Correct references:

Nguyen T.T.H., H.T. Nguyen, S.P.H. Tong, L. Nguyen-Ngoc, 2019a. Vegetation Biomass of Sargassum Meadows in An Chan Coastal Waters, Phu Yen Province, Vietnam Derived from PlanetScope Image. Journal of Environmental Science and Engineering 8:81–92.

Nguyen T.T.H., H.T. Nguyen, V.T Nguyen, Lam Nguyen-Ngoc, 2019b. Spatial distribution of submerged aquatic vegetation in An Chan coastal waters, Phu Yen province using the PlanetScope satellite image. Vietnam Journal of Earth Sciences 41:358–373.

Nguyen V.L. 2009. Coral reef fishes in the coastal waters of south-central Vietnam. Vietnam Journal of Marine Science and Technology 9.

Nguyen V.LL, Phan HK. 2008. Distribution and factors influencing on structure of reef fish communities in Nha Trang Bay Marine Protected Area, South-Central Vietnam. Environmental biology of fishes 82:309–324.

Annotated reviews are not available for download in order to protect the identity of reviewers who chose to remain anonymous.

---

## Round 0.3 · accepted · Accept

The reviewers' comments have been addressed in the revision. There are no more concerns about the quality of the manuscript. Congratulations.